# Hierarchical Cross Contrastive Learning of Visual Representations

## Abstract

The rapid progress of self-supervised learning (SSL) has greatly reduced the labeling cost in computer vision. The key idea of SSL is to learn invariant visual representations by maximizing the similarity between different views of the same input image. In most SSL methods, the representation invariant is measured by a contrastive loss which compares one of the network outputs after the projection head to its augmented version. Albeit being effective, this approach overlooks the information containing in the hidden layer of the projection head therefore could be sub-optimal. In this work, we propose a novel approach termed Hierarchical Cross Contrastive Learning(HCCL) to further distill the information mismatched by the conventional contrastive loss. The HCCL uses a hierarchical projection head to project the raw representations of the backbone into multiple latent spaces and then compares latent features across different levels and different views. By cross-level contrastive learning, HCCL not only regulates invariant on multiple hidden levels but also crosses different levels, improving the generalization ability of the learned visual representations. As a simple and generic method, HCCL can be applied to different SSL frameworks. We validate the efficacy of HCCL under classification, detection, segmentation, and few-shot learning tasks. Extensive experimental results show that HCCL outperforms most previous methods in various benchmark datasets.

## 1 Introduction

Self-supervised learning (SSL) of visual representation aims to learn representations from image pixels without semantic labels. Recently, contrastive representation learning has gained growing attention by demonstrating state-of-the-art performance in SSL for large-scale image recognition, closing the gap to supervised baselines (Chen et al., 2020; Grill et al., 2020; Chen & He, 2021). Contrastive representation learning is essentially an instance-level discrimination task. It narrows the distance of the representation generated by the same image, as an instance class, while pushing away the representation generated by other images to learn invariant visual representations. In most contrastive learning methods, the representation invariant is measured by a contrastive loss, which compares one of the network outputs after the projection head to its augmented version. Some works attempt to explore advanced training methods to keep the contrastive loss stable in SSL, by clustering, memory bank, momentum update or projector head(Caron et al., 2020; He et al., 2020; Grill et al., 2020). Some other works seek to improve representation performance by increasing transformation views with complicated augmentations. In this paper, we focus on improving contrastive learning by studying the projection head.

As shown above, the projection head serves as a bridge to squeeze out redundant information in order to retrain invariant information from the backbone for the contrastive loss module. However, only the last output of the projection head is used, while hidden layers are overlooked, which loses many valuable information according to the information bottleneck theory(Tishby et al., 2000; Tishby & Zaslavsky, 2015). Inspired by the successful results of knowledge distillation in supervised representation learning, we propose a cross-level contrastive learning method for SSL(Sun et al., 2019). In this framework, contrastive losses are applied in the intermediate layers of projection heads. In particular, a cross-layer assignment is adapted to create pairs between different views, as the representation generated at the same level of projection heads of different transformed views is likely to retrain similar information, which is not conducive to learning extra knowledge through contrast-

ing. The consistency assumption over the representation in different views and different layers has been partially explored in many feature regularization approaches(Guo et al., 2020; Hou et al., 2019; Wang et al., 2019; Huang et al., 2020). This phenomenon is investigated in BYOL, which shows that the use of different projection heads for contrastive learning can achieve better performance.

In this paper, a series of experiments and analyses are proposed to validate our contributions. The experimental results show the effectiveness of our method in improving the robustness of the pre-training model for downstream tasks. Specifically, in the ImageNet(Deng et al., 2009) linear evaluation task, the standard ResNet-50(He et al., 2016) trained by our method achieves 73.5% top-1 accuracy. In the object detection and instance segmentation tasks, our method also outperforms random initialization and existing image-level self-supervision methods on PASCAL VOC(Everingham et al., 2010) and COCO(Lin et al., 2014) datasets. Last but not the least, compared to other SSL methods, HCCL requires fewer training epoch to learn effective representations. For example, it only takes 100 epochs to achieve 69.5% top-1 accuracy in the ImageNet linear evaluation task, while other methods may take 200 epochs or more. Our main contributions can be summarized as follows:

- In this paper, we propose a Hierarchical Cross Contrastive Learning (HCCL) method to explore information from hidden layers of the projection head. Experiments show that our method can achieve state-of-the-art results in a series of standard benchmarks.
- Ablation studies are proposed to evaluate the modules of CLIC. Experiments show that cross-level assignments, multi-layer outputs and multi-predictors support the final performance improvements.
- We evaluate the generalization of HCCL method in both SimSiam and BYOL frameworks. The experiments show that HCCL can significantly improve performance and speed up convergence.

## 2 RELATED WORK

Self-supervised learning or unsupervised visual representation learning has undergone a relatively long period of development. Early works use generative methods to restore the distribution of input data, usually including auto-encoders (Kingma & Welling, 2013; Vincent et al., 2008) and adversarial learning (Goodfellow et al., 2014; Donahue et al., 2016; Dumoulin et al., 2016). However, these methods are intended to restore the high-level details of the image that are not necessary for downstream tasks. On the other hand, pixel-level reconstruction also costs a lot of calculations.

Some other methods learn specific representations by handcrafted pretext tasks. These pretext tasks include colorization(Zhang et al., 2016; Larsson et al., 2016), jigsaw puzzle(Noroozi & Favaro, 2016; Misra et al., 2016), relative patch prediction (Doersch et al., 2015), inpainting(Pathak et al., 2016), geometric transformations(Dosovitskiy et al., 2014), etc. However, the design of excuse tasks relies on strong prior knowledge, and the representations learned by these methods usually lack generality in downstream tasks.

Another type of works is the clustering based self-training (Bautista et al., 2016; Asano et al., 2019; Caron et al., 2018; 2019; 2020). In general, they cluster the representations and learn to predict the cluster assignment. DeepCluster (Caron et al., 2018; 2019) uses k-means to create pseudo labels for samples. It clusters data using previous versions of representation and uses the cluster index of each sample as a classification target for the new representation. SwAV calculates the assignment from one view and predicts it from another (Caron et al., 2020). It produces soft assignment by the Sinkhorn-Knopp algorithm (Cuturi, 2013) and performs online clustering under a balanced partition constraint for each batch. Clustering-based methods require a costly clustering phase to avoid collapsing, and a large memory bank or large batches to provide sufficient samples for clustering.

Contrastive learning measures the similarity of pairs of samples in the representation space by using contrast loss(Hadsell et al., 2006). It brings the representation of different views of the same sample closer and pushes the representation of views from different samples away. Dosovitskiy et al. (2014) considers each image in a dataset as its own class represented by a feature vector and separates each image. Since this method produces a huge classification header that is difficult to handle, Wu et al. (2018) use a memory bank that stores previously-computed instance representation vectors to replace the classifier. Several later works also follow and extend this approach(Misra & Maaten,

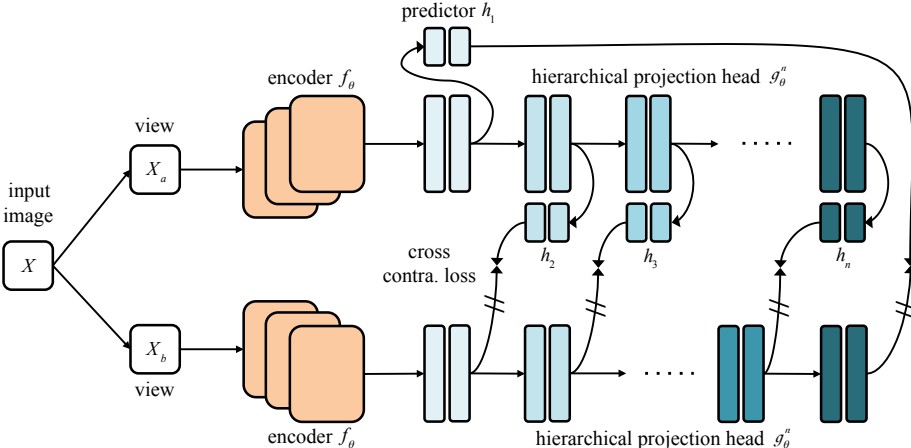

Figure 1: **HCCL's framework.** HCCL minimizes the cross contrastive loss between the different level output of hierarchical projection head. The encoder parameters and projection parameters of the two branches are shared. '=' denotes stop gradient.

2020; Tian et al., 2020; Zhuang et al., 2019). MoCo(He et al., 2020) reduces the size of the memory bank by maintaining a dynamic queue that stores negative sample representations. The representation in the queue is generated by a moving average encoder to improve consistency with the query representation. SimCLR(Chen et al., 2020) removes the memory bank and uses the negative samples that exist in the current batch directly for learning. It introduces nonlinear projection heads and complex composition of image transformations to improve the robustness of representations in downstream tasks. BYOL(Grill et al., 2020) further examines the need for negative samples, using a slow-moving average target network to produce stable targets for the online network. The online network bootstraps its own representation by keeping pace with stable targets. By analysing certain conditions for avoiding collapsing solutions, SimSiam(Chen & He, 2021) removed the momentum encoder on the basis of BYOL. BARLOW TWINS(Zbontar et al., 2021) proposes an objective function that naturally avoids collapse by measuring the cross-correlation matrix between the outputs of two siamese networks fed with different views of a sample and bringing it as close as possible to the identity matrix. It causes the representation of different views of a sample to be similar, while minimizing the redundancy between the components of these representations. The above-mentioned self-supervised learning methods all express learning as image-level predictions using global representations, and the gap between image-level pre-training and dense prediction tasks still exists. Some works propose pixel-level self-supervised learning methods to improve the effect of pre-training representations on dense prediction tasks(Wang et al., 2021; Roh et al., 2021; Xie et al., 2021; Pinheiro et al., 2020).

## 3 METHOD

In this section, we describe the proposed HCCL in detail. First, we briefly introduce the pipeline of the proposed HCCL shown in Figure 1, including the design of a hierarchical projection network that maps representations to different latent spaces. Then we introduce the cross contradictive loss, which allows explicit information interactions between different latent layers. Finally, we present the implementation details in HCCL, such as image transforms, architecture, and optimization.

### 3.1 FRAMEWORK

HCCL builds on the cross-view prediction framework introduced in recent successful contrastive learning approaches(Chen et al., 2020; Grill et al., 2020; Chen & He, 2021). The cross-view prediction framework is based on a prediction problem that the representation of a transformed view of an image should be predictive of the representation of another transformed view of the same image. HCCL transforms an image $X$ randomly to two distorted views $X_a$ and $X_b$, by applying transformations sampled from the set $\mathbb{T}$ of image transformations. The two distorted views $X_a$ and $X_b$

are then fed to non-linear mapping encoder $f_\theta$, typically a deep network with trainable parameters $\theta$, producing two raw representations $\boldsymbol{y}_a$ and $\boldsymbol{y}_b$ respectively. For $f_\theta$, HCCL allows various choices of the network architecture without any constraints. $\boldsymbol{y}_a$ and $\boldsymbol{y}_b$ are representations used for transfer tasks. After extracting the raw representation $\boldsymbol{y}_a$ and $\boldsymbol{y}_b$, a hierarchical neural network projection head $g_\theta^n$ that maps them to different latent space, produces $n$ embedding $\mathbb{Z}_a^n = \{\boldsymbol{z}_{a1}, \boldsymbol{z}_{a2}, \dots, \boldsymbol{z}_{an}\}$ and $\mathbb{Z}_b^n = \{\boldsymbol{z}_{b1}, \boldsymbol{z}_{b2}, \dots, \boldsymbol{z}_{bn}\}$. Further, $n$ prediction MLP head, denoted as $h_1, h_2, \dots, h_n$, transforms the projection embedding $\mathbb{Z}_a^n$ and $\mathbb{Z}_b^n$ to $2n$ vectors as $\mathbb{P}_a^n = \{\boldsymbol{p}_{a1}, \boldsymbol{p}_{a2}, \dots, \boldsymbol{p}_{an}\}$ and $\mathbb{P}_b^n = \{\boldsymbol{p}_{b1}, \boldsymbol{p}_{b2}, \dots, \boldsymbol{p}_{bn}\}$. The following formulas summarizes the above pipeline:

$$\mathbb{Z}_a^n = g_\theta^n(f_\theta(\boldsymbol{X}_a)) \tag{1}$$

$$\mathbb{Z}_b^n = g_\theta^n(f_\theta(\boldsymbol{X}_b)) \tag{2}$$

$$\boldsymbol{p}_{ai} = h_i(\boldsymbol{z}_{ai}) \tag{3}$$

$$\boldsymbol{p}_{bi} = h_i(\boldsymbol{z}_{bi}) \tag{4}$$

where $\boldsymbol{y}_a = f_\theta(\boldsymbol{X}_a)$, $\boldsymbol{y}_b = f_\theta(\boldsymbol{X}_b)$ and $i \in \{1, 2, \dots, n\}$. Follow the same way as SimSiam(Chen & He, 2021) and BYOL(Grill et al., 2020), HCCL matches one view's prediction vector to the other view's projection embedding finally. The difference is that we have multiple projection embedding and prediction vectors, and cannot directly use the original contrastive loss, so we designed a cross contrastive loss to solve it.

## 3.2 Cross Contrastive Loss

Since $\mathbb{Z}^n$ are extracted from different levels of the projection head, the high-level representations are more invariant and the low-level representations are more informative. HCCL adopts an intuitive approach, that is, cross contrastive representations of different levels to trade off invariant and informative. Given the projection embedding $\boldsymbol{z}$ and the prediction vector $\boldsymbol{p}$, we minimize their mean squared error of $l_2$-normalized vectors:

$$D(\boldsymbol{p}, \boldsymbol{z}) = -\frac{\boldsymbol{p} \cdot \boldsymbol{z}}{||\boldsymbol{p}||_2 \cdot ||\boldsymbol{z}||_2} \tag{5}$$

where $|| \cdot ||_2$ is $l_2$-norm. After that, we define the cross contrastive loss as:

$$L_{cross}(\mathbb{P}^n, \mathbb{Z}^n) = D(\boldsymbol{p}_1, \boldsymbol{z}_n) + \sum_{i=2}^{n} D(\boldsymbol{p}_i, \boldsymbol{z}_{i-1}) \tag{6}$$

Here each level of prediction vector contrast with the previous level of projection representation except the first level one. The first level of the prediction vector contrast with the final level of projection representation. Following BYOL and SimSiam, We symmetrize the loss $L_{cross}$ in Eq.6 and take a stop-gradient operation to projection embedding $\mathbb{Z}^n$. The form of total loss is implemented as:

$$L = L_{cross}(\mathbb{P}_a^n, stopgrad(\mathbb{Z}_b^n)) + L_{cross}(\mathbb{P}_b^n, stopgrad(\mathbb{Z}_a^n)) \tag{7}$$

Here the encoder only receives gradient from branch $\mathbb{P}_a$ and $\mathbb{P}_b$ in the two terms. The pseudo-code is provided in Algorithm refalgorithm

## 3.3 Implementation Details

**Image augmentations** We use the same image augmentation set and parameters as BYOL(Grill et al., 2020). Firstly, two different views are randomly cropped from each input image and resized to $224 \times 224$, followed by horizontal flipping, color distortion, converting to grayscale, Gaussian blurring, and polarization. The color distortion consists of a random sequence of brightness, contrast, saturation, hue adjustments. Random cropping and resizing are always applied, others are applied randomly with a certain probability. For gaussian blurring and polarization, the probabilities applied on the two distorted views are different.

**Architecture** We use ResNet-50(He et al., 2016) as our base parametric encoder network, the fully-connected layer after global average pooling is removed and replaced by a hierarchical projection network. Specifically, We adopt a 2-level projector network that projects the output of the global average pooling layer to two latent spaces. Each level projector network has three linear layers and each layer has 2048 output units. The first two linear layers of each level projector are followed

---

**Algorithm 1** PyTorch-style pseudocode for HCCL

---

```
# f: encoder networks
# g: hierarchical projetcion head(with two levels)
# h1, h2: two prediction networks
# n: batch size

for x in loader: # load a minibatch x with n samples
    x_a, x_b = augment(x)               # two augmented views
    y_a, y_b = f(x_a), f(x_b)           # encoder output
    z_a1, z_a2 = g(y_a)                 # proj outputs of view a
    z_b1, z_b2 = g(y_b)                 # proj outputs of view b
    p_a1, p_a2 = h1(z_a1), h2(z_a2)     # pred outputs of view a
    p_b1, p_b2 = h1(z_b1), h2(z_b2)     # pred outputs of view b
    loss = [L(p_a1, z_b2)/2 + L(p_a2, z_b1) +  \\
            L(p_b1, z_a2)/2 + L(p_b2, z_a1)]/2
    loss.backward()                     # back-propagate
    optimizer.step()

def L(t, s):
    s = s.detach()          # stop gradient
    t = normalize(t)
    s = normalize(s)
    return -(t * s).sum(dim=1).mean()
```

---

by a batch normalization layer and rectified linear units(ReLU). Each level projector network is followed by a prediction network which has two linear layers. The dimension of the prediction network's input and output is 2048 and the hidden layer's dimension is 512. The prediction network has batch normalization and ReLU applied to its hidden layer. Its output layer does not have batch normalization or ReLU.

**Optimization** We adapt the same optimization protocol described in SimSiam(Chen et al., 2020). We use SGD optimizer(You et al., 2017) with a momentum parameter of 0.9. We use cosine decay learning rate schedule(Loshchilov & Hutter, 2016) and train for 800 epochs with a batch size of 512, without restarts and warm-up. We set the initial learning rate to 0.025 and multiply the learning rate by the batch size and divide it by 256(Goyal et al., 2017). In addition, we use a global weight decay parameter of 1.5·10-4 while excluding the biases and batch normalization parameters from weights. Following SimCLR(Chen et al., 2020) We use batch normalization synchronized across devices. Training is distributed across 8 V100 GPUs and takes approximately 230 hours for a ResNet-50. For 100, 200, 400, and other less epoch training, we set the initial learning rate to 0.05 and weight decay parameter to 1·10-4. Since SimSiam(Chen et al., 2020) and BYOL(Grill et al., 2020) have explained that the predictor network should adapt to the latest representation, so it is not necessary to force it to converge before the representations. In our experiment, we keep the learning rate of one predictor constant and set the learning rate of the other predictor to 10 times the initial learning rate and decay normally.

## 4 EXPERIMENTS

In this section, we analyze the performance of HCCL's representations after self-supervised pre-training on the training set of the ImageNet ILSVRC-2012 dataset(Deng et al., 2009) by transfer learning on different datasets and tasks. Firstly, we evaluate the pre-trained representation on ImageNet in both linear classification on frozen features and semi-supervised learning. Then we assess its transfer capabilities on other vision datasets and tasks, including classification, segmentation, and object detection.

Table 1: **Top-1 accuracy under linear evaluation on ImageNet.** All methods are based on ResNet-50 pre-trained with two 224×224 views. Evaluation is on a single crop and all best results are in **bold**. '†' denotes improved reproduction from SimSiam.

| Method | Encoder | momentum encoder | 100ep | 200ep | 300ep | 400ep | 800ep |
|---|---|---|---|---|---|---|---|
| SimCLR† | R50 | | 66.5 | 68.3 | - | 69.8 | 70.4 |
| SWAV† | R50 | | 66.5 | 69.1 | - | 70.7 | 71.8 |
| Barlow Twins | R50 | | - | - | 71.4 | - | 73.2 |
| MoCoV2† | R50 | $\checkmark$ | 67.4 | 69.9 | - | 71.0 | 72.2 |
| BYOL† | R50 | $\checkmark$ | 66.5 | 70.6 | **72.5** | **73.2** | **74.3** |
| DINO | ViT-S | $\checkmark$ | 67.8 | - | **72.5** | - | - |
| SimSiam | R50 | | 68.1 | 70.0 | - | 70.8 | 71.3 |
| **HCCL** | R50 | | **69.5** | **71.4** | - | 72.4 | 73.5 |

## 4.1 LINEAR AND SEMI-SUPERVISED EVALUATIONS ON IMAGENET

**Linear evaluation on ImageNet** We first verify our method by training a linear classifier on ImageNet on fixed representations, which are from ResNet's global average pooling layer. Following a common protocol, the linear classifier is trained for 90 epochs with a batch size of 1024 and a cosine learning rate schedule. We set the initial learning rate to 0.1 and multiply the learning rate by the batch size and divide it by 256. We minimize the cross-entropy loss with the LARS optimizer with a momentum of 0.9 and weight decay of 0. At training time, the input image is randomly cropped, resized to 224×224, and randomly flipped horizontally. At test time, the image is resized to 256×256 and center-cropped to a size of 224×224.

We compare with the state-of-the-art frameworks in Table 1 on ImageNet linear evaluation. All competitors are based on original papers or reproduction trained in SimSiam(Chen & He, 2021). All competitors are based on a standard ResNet50, with two 224×224 views used during pre-training. Table 1 reports the top-1 accuracies obtained on the ImageNet validation set and the main properties of the methods. HCCL achieves competitive results with a top-1 accuracy of 73.5%, which is comparable to the state-of-the-art methods. HCCL is trained with small batch size and used neither negative samples nor a momentum encoder. It achieved the best results among all methods under 100-epoch and 200-epoch pre-training. HCCL only 0.8% below the performance of BYOL(Grill et al., 2020). BYOL uses a momentum encoder, which means a greater training cost, because it requires 4 times forward in one iteration, while HCCL only requires 2 times. In addition, HCCL can also be flexibly applied to the SSL method with momentum encoder. We evaluated HCCL with momentum encoder, it achieved better performance than BYOL.(See Appendix A.2)

**Semi-supervised learning on ImageNet** We simply fine-tune the whole ResNet-50 pre-trained with our method on a small subset of ImageNet's train set and evaluate the performance. We sample

Table 2: **Semi-supervised learning on ImageNet.** All models are finetuned with 1% and 10% training examples. Results for the supervised method are from Zhai et al. (2019).

| Method | TOP-1 | | TOP-5 | |
|---|---|---|---|---|
| | 1% | 10% | 1% | 10% |
| Supervised | 25.4 | 56.4 | 48.4 | 80.4 |
| PIRL | - | - | 57.2 | 83.8 |
| SimCLR | 48.3 | 65.6 | 75.5 | 87.8 |
| BYOL | 53.2 | 68.8 | 78.4 | 89.0 |
| SWAV | 53.9 | 70.2 | 78.5 | 89.9 |
| Barlow Twins | 55.0 | 69.7 | 79.2 | 89.3 |
| **HCCL** | **56.4** | **71.3** | **80.1** | **90.4** |

Table 3: **Transfer learning on image classification task.** We report top-1 accuracy on iNat18 and Places-205 datasets, and classification mAP on VOC07.

| Method | iNat18 | Place-205 | VOC07 |
|---|---|---|---|
| Supervised | 46.7 | 53.2 | 87.5 |
| PIRL | 29.7 | 51.0 | 78.8 |
| SimCLR | 37.2 | 52.5 | 85.5 |
| MoCov2 | 38.6 | 51.8 | 86.4 |
| SWAV | 39.5 | 52.8 | 86.4 |
| BYOL | **47.6** | 54.0 | **86.6** |
| Barlow Twins | 46.5 | 54.1 | 86.2 |
| **HCCL** | 47.2 | **54.5** | **86.6** |

Table 4: **Transfer Learning on object detection and instance segmentation task.** All models use the C4-backbone. We benchmark finetuned representations on the object detection task on VOC07+12 using Faster R-CNN and on the detection and instance segmentation task on COCO using Mask R-CNN(1× schedule).

| Method | VOC 07+12 det | | | COCO Det | | | COCO instance seg | | |
|---|---|---|---|---|---|---|---|---|---|
| | $AP_{50}$ | $AP_{75}$ | AP | $AP_{50}$ | $AP_{75}$ | AP | $AP_{50}^m$ | $AP_{75}^m$ | $AP^m$ |
| Random Init. | 59.0 | 31.6 | 32.8 | 50.9 | 35.3 | 32.8 | 47.9 | 32.0 | 29.9 |
| Supervised | 81.3 | 58.8 | 53.5 | 58.2 | 41.2 | 38.2 | 54.7 | 35.2 | 33.3 |
| SimCLR | 81.8 | 61.4 | 55.5 | 57.7 | 40.9 | 37.9 | 54.6 | 35.3 | 33.3 |
| MoCoV2 | 82.3 | 63.3 | 57.0 | 58.8 | **42.5** | 39.2 | 55.5 | 36.6 | 34.3 |
| BYOL | 81.4 | 61.1 | 55.3 | 57.8 | 40.9 | 37.9 | 54.3 | 35.0 | 33.2 |
| SWAV | 81.5 | 61.4 | 55.4 | 57.6 | 40.3 | 37.6 | 54.2 | 35.1 | 33.1 |
| SimSiam | 82.4 | 63.7 | 57.0 | 59.3 | 42.1 | 39.2 | 56.0 | **36.7** | 34.4 |
| Barlow Twins | **82.6** | 63.4 | 56.8 | 59.0 | **42.5** | 39.2 | 56.0 | 36.5 | 34.3 |
| HCCL | **82.6** | **64.3** | **57.2** | **60.1** | 42.3 | **39.5** | **56.5** | **36.7** | **34.6** |

1% and 10% of the labeled ImageNet training datasets in a class-balanced way by following Sim-CLR(Chen et al., 2020). The image augmentations are the same as in the linear evaluation setting. We train for 20 epochs with a batch size of 256 and a momentum of 0.9. We minimize the cross-entropy loss with the SGD optimizer with Nesterov momentum and do not use any regularization methods. We use a learning rate of 0.02 for the conv-net weights and 300 for the final linear layer, and we decay the learning rates with a cosine learning rate schedule. Table 2 shows the comparisons of our semi-supervised results against recent methods. We report both top-1 and top-5 accuracies on the ImageNet validation set. HCCL is either on par or slightly outperforms previous competing semi-supervised and semi-supervised approaches.

## 4.2 TRANSFER TO OTHER DATASETS AND TASKS

**Image classification with fixed features** We follow the exact settings from PIRL(Misra & Maaten, 2020) for training and evaluating linear classifiers on the iNaturalist2018(Van Horn et al., 2018), Places205(Zhou et al., 2014) and VOC07(Everingham et al., 2010) datasets. For VOC07, We train a linear SVM on the global average pooled final representations. For iNaturalist2018 and Places-205, we train a linear classifier with SGD using a learning rate of 0.01 reduced by a factor of 10 at two equally spaced intervals, a batch size of 256, a weight decay of 0.0001, and SGD momentum of 0.9. We train the linear models for 30 epochs on Places-205 and 84 epochs on iNat18. We report the top-1 accuracy computed using the 224 × 224 center crop on the validation set. The results in Table 3 show that HCCL almost outperforms all prior works on three datasets.

**Object Detection and Instance Segmentation** We evaluate our representations by transferring them to localization-based tasks, like object detection and instance segmentation. Following the setup in MOCO(He et al., 2020), we fine-tune all layers of the pre-trained models end-to-end in the VOC07+12(Everingham et al., 2010) and COCO(Lin et al., 2014) datasets. We use Faster R-CNN(Ren et al., 2015) with an R50-C4 backbone on VOC and use Mask R-CNN(He et al., 2017) with an R50-C4 backbone on COCO.

We use the detection models as implemented in Detectron2 library(Wu et al., 2019) for training and closely follow the finetune and evaluation settings from the public codebase of MoCo. For Pascal VOC, We use the VOC07+12 trainval set for training, and report results on the VOC07 test set averaged over 5 independent runs. We train a Faster R-CNN C4 backbone for 24K iterations using a batch size of 16 across 8 GPUs using SyncBatchNorm. We set the initial learning rate to 0.1 and reduce it by a factor of 10 after 18K and 22K iterations. We use linear warmup(Goyal et al., 2017) with a slope of 0.333 for 1000 iterations. For COCO, We use the COCO 2017 train split for training and report results on the val split averaged over 5 independent runs. We set the initial learning rate to 0.03 and keep the other parameters the same as in the 1× schedule in MoCo's codebase.

In Table4, we show that HCCL outperforms among these leading methods on both Pascal VOC and COCO datasets. It means that HCCL's representations are transferable beyond the classification task.

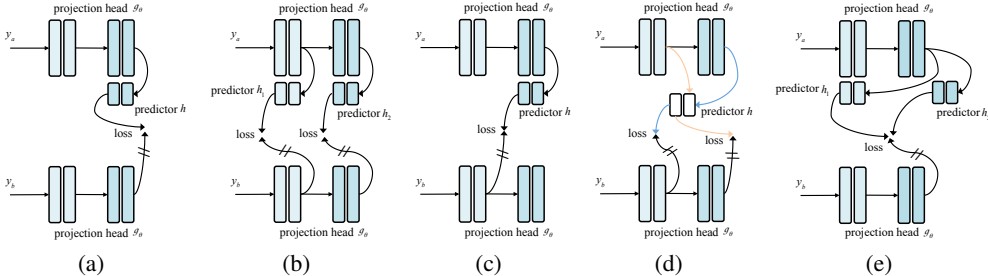

(a)      (b)      (c)      (d)      (e)

Figure 2: **Comparison of different projection architectures.** The projection head includes all layers that can be shared between both branches. '=' denotes stop gradient.

## 5 ABLATIONS

We present extensive ablation experiments on HCCL to give an intuition of its behavior and performance and show how each component contributes. We train HCCL for 100 epochs among all ablation experiments, which performs consistent results compared to our baseline training of long epochs. For reproducibility, all the experiment results are averaged over 3 independent trials. For all the experiments in this section, we set the initial learning rate to 0.05 with batch size 512, the weight decay to 10-4 as in SimSiam(Chen et al., 2020). We report the top-1 accuracy of training linear classifiers on the ImageNet under the linear evaluation protocol the same as in Section 4.

### 5.1 INFLUENCE OF HIERARCHICAL PROJECTION HEAD AND CROSS CONTRASTIVE LOSS

In this subsection, we alter our hierarchical projection head and cross contrastive loss in several ways to test the necessity of each term of them. Figure 2 recapitulates the different structure and learning ways on projection head and Table 5 shows their results on a linear evaluation benchmark of Imagenet.

**Deep projection with single level** We first confirm that the gain of HCCL is not brought about by a deep projection head with a single level. In Figure 2(a), we use a 6 layer projection head the same as in section 4, only single-level representation (the output of the final linear layer) are used to construct the contrastive loss. Compared to HCCL, the performance drops by 0.8%, which means that the lack of hierarchical representation and cross contrastive loss will cause performance degradation.

**Hierarchical projection head without cross contrastive loss** In Figure 2(b), we use a 6 layer projection head the same as in section 4, and multi-level representation (the output of the final linear layer and the 3rd linear layer) are used to construct the contrastive loss. Instead of cross contrastive between different levels, we contrast representations of the same level with each other. Compared with HCCL the performance reduced by 0.8%, which means that hierarchical representations must be combined with a cross contrastive loss to improve performance.

**Hierarchical projection head with single cross contrastive loss** Figure 2(c) and Figure 2(b) use the same hierarchical projection structure, the difference is that in Figure 2(c) we use the high-level feature of view1 and the low-level feature of view2 for cross contrastive. Compared with the

Table 5: **Influence of hierarchical projection head and cross contrastive loss.** We report the top-1 accuracy of models trained with different choices on hierarchical projection head, cross contrastive loss, and multi predictor.

| Hierarchical | Cross Contrastive Loss | Multi Predictor | Top-1 Acc |
|:---:|:---:|:---:|:---:|
| | | | 68.6 |
| √ | | √ | 68.6 |
| | √ | | 68.4 |
| √ | √ | | 68.7 |
| | | √ | 68.4 |
| √ | √ | √ | **69.5** |

Table 6: **The levels and layers of projection head.** We examine the effect of different levels and layers(in one level) for the hierarchical projection head. We do not apply a ReLU activation nor a batch normalization on the final linear layer of our MLPs.

|  |  | Layers in one level | | |
|---|---|---|---|---|
|  |  | 2 | 3 | 4 |
|  | 2 | 68.4 | **69.5** | 68.8 |
| **Levels** | 3 | 69.0 | 69.1 | 68.2 |
|  | 4 | 69.0 | 68.9 | 67.7 |

Table 7: **Learning rate of Predictor.** We report top-1 accuracy at 100 epochs when applying a multiplier $\lambda_l$ to the low-level predictor and $\lambda_h$ to the high-level predictor learning rate. 'fixed' denotes using constant learning rate.

| $\lambda_l$ |  | $\lambda_h$ | | | | |
|---|---|---|---|---|---|---|
|  |  | Fixed | 5 | 10 | 20 | 40 |
|  | Fixed | 68.3 | 69.2 | 69.4 | 69.2 | 69.4 |
|  | 5 | 68.9 | 69.1 | 69.3 | 69.2 | **69.5** |
| $\lambda_l$ | 10 | 69.4 | 69.4 | 69.3 | 69.3 | **69.5** |
|  | 20 | **69.5** | 69.2 | 69.4 | 69.1 | 69.3 |
|  | 40 | 69.3 | 69.4 | 69.4 | **69.5** |  |

standard HCCL structure, the cross contrastive between the low-level feature of view1 and the high-level of view2 is removed. It brings 1.0% drops compared with HCCL. Obviously that only a one-way contrastive between high-level and low-level representations cannot improve the performance.

**Hierarchical projection head with single predictor** In Figure 2(d), we share a single predictor among all level projections, which brings 0.7% drops. The output representations of different levels of projection heads are distributed in different spaces, and the use of a shared single predictor will destroy this distribution and lead to worse performance.

**Deep projection with multi predictor** We confirm that the gain of HCCL is not brought about by multi predictor. In Figure 2(e), we use a 6 layer projection head the same as in section 4. The output of the projection head is fed to two different predictors, and then the contrastive loss is constructed. Compared to HCCL, the performance drops by 1.1%.

## 5.2 LEVELS AND DEPTH OF PROJECTOR NETWORK

Table 6 shows the influence of projector architecture on HCCL. We examine the effect of different levels and depths for the projector network. Using the default projector network of level 2 and depth 3 yields the best performance. In addition, we find that our model performs worse when the projector network has more layers, with a saturation of the performance for 3 layers and 2 levels.

## 5.3 PREDICTOR LEARNING RATE

In this subsection, we examine the effect of the combination of learning rates used by different predictors in HCCL. BYOL(Grill et al., 2020) and SimSiam(Chen & He, 2021) explains that predictors should adapt to the latest representations and keep near-optimal, so use a sufficiently large learning rate of predictor provides a reasonably good performance. We use the default HCCL structure of level 2 and depth 3, with 2 different predictors. We multiply the learning rate of the predictors by a constant $\lambda$ compared to the learning rate used for the rest of the network; all other hyper-parameters are unchanged. Table 7 provides the performance of the two predictors using constant learning rate or different $\lambda$ times learning rate. Results show that the predictor learning rate needs to be much higher than the projector learning rate.

## 6 CONCLUSION

We propose HCCL, a simple yet effective method strategy that applies to a class of self-supervised learning. The main idea of HCCL is to explore multiple outputs from hidden layers instead of the output of the last one, and to compare latent features on different levels and views. We show that HCCL regulates not only invariant on multiple hidden levels, but also across different levels, improving the generalization ability of the learned visual representations. We also show that HCCL can work in various SSL frameworks like BYOL and Simsiam. The results of the experiments suggest that HCCL is particularly effective when computing resources are limited, which is widespread in practice.

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

# A APPENDIX

## A.1 THE PROOF OF HCCL'S CONVERGENCE

We consider a deep linear system where $\boldsymbol{x} \in \mathbb{R}^d$ is the input image, $\boldsymbol{W} \in \mathbb{R}^{m \times d}$ extract $d$-dimensional latent features from $\boldsymbol{x}$. The augmented version of $\boldsymbol{x}$ is $\hat{\boldsymbol{x}}$. The fc layer $\boldsymbol{w}_{\text{fc}}$ generates the final label $y = \boldsymbol{w}_{\text{fc}}^\top \boldsymbol{W} \boldsymbol{x}$. The ground-truth is denoted as $y^* = \boldsymbol{w}_{\text{fc}}^{*\top} \boldsymbol{W}^* \boldsymbol{x}$. The matrix $\hat{\boldsymbol{W}}_1 \in \mathbb{R}^{m \times d}$ denotes the stopping-gradient copy of $\boldsymbol{W}_1 \in \mathbb{R}^{m \times d}$, $\hat{\boldsymbol{W}}_1 = \text{StopGrad}(\boldsymbol{W}_1)$. Similary, $\boldsymbol{W}_2, \hat{\boldsymbol{W}}_2, \boldsymbol{P}_1, \boldsymbol{P}_2, \hat{\boldsymbol{P}}_1, \hat{\boldsymbol{P}}_2 \in \mathbb{R}^{m \times m}$. The learning rates for $\{\boldsymbol{W}_1, \boldsymbol{W}_2\}$ and $\{\boldsymbol{P}_1 \boldsymbol{P}_2\}$ are $\{\eta_w, \eta_p\}$ respectively.

With out loss of generality, we assume the features of $\boldsymbol{x}$ can be divided into two parts $\boldsymbol{x} = [\boldsymbol{x}_1, \boldsymbol{x}_2]$ where $\boldsymbol{x}_1$ corresponds to the augmentation invariant part and $\boldsymbol{x}_2$ corresponds to the augmentation parts. Similarly, $\hat{\boldsymbol{x}} = [\boldsymbol{x}_1; \hat{\boldsymbol{x}}_2]$. We futher assume that the invariant part and augmentation part are independent and the augmentation is i.i.d. generated, that is, $\mathbb{E}\{\boldsymbol{x}\boldsymbol{x}^\top\} = \boldsymbol{I}_{d \times d}$, $\mathbb{E}\{\boldsymbol{x}_1\boldsymbol{x}_1^\top\} = \boldsymbol{I}_{m \times m}$, $\mathbb{E}\{\boldsymbol{x}_2\boldsymbol{x}_2^\top\} = \boldsymbol{I}_{(d-m) \times (d-m)}$, $\mathbb{E}\{\boldsymbol{x}_1\boldsymbol{x}_2^\top\} = \boldsymbol{0}$, $\mathbb{E}\{\boldsymbol{x}_1\hat{\boldsymbol{x}}_2^\top\} = \boldsymbol{0}$, $\mathbb{E}\{\boldsymbol{x}_2\hat{\boldsymbol{x}}_2^\top\} = \boldsymbol{0}$. Aligned with $\boldsymbol{x}$, matrix $\boldsymbol{W}$ can be divided into two parts too, $\boldsymbol{W} = [\boldsymbol{U}; \boldsymbol{V}]$. Ideally, $\|\boldsymbol{U}^*\| \gg 0$ and $\|\boldsymbol{V}^*\| = \boldsymbol{0}$ gives the optimal solution. The measure the quality of self-supervised learning, we define $\rho = \|\boldsymbol{V}\|/\|\boldsymbol{U}\|$ as the recovering error of invariant feature subspace. $\rho = 0$ means that the SSL algorithm finds the perfect $\boldsymbol{W}$ without lossing any userful information.

In our method, the loss function is

$$\mathcal{L} = \mathbb{E}_{\boldsymbol{x},\hat{\boldsymbol{x}}} \left\{ \frac{1}{2} \| \boldsymbol{P}_1 \boldsymbol{W}_1 \boldsymbol{x} - \hat{\boldsymbol{W}}_2 \hat{\boldsymbol{W}}_1 \hat{\boldsymbol{x}} \|^2 + \frac{1}{2} \| \boldsymbol{P}_2 \boldsymbol{W}_2 \boldsymbol{W}_1 \boldsymbol{x} - \hat{\boldsymbol{W}}_1 \hat{\boldsymbol{x}} \|^2 \right\}$$

The gradients are

$$\begin{aligned}
\nabla_{P1}\mathcal{L} &= \mathbb{E}_{\boldsymbol{x},\hat{\boldsymbol{x}}} \left\{ (\boldsymbol{P}_1 \boldsymbol{W}_1 \boldsymbol{x} - \hat{\boldsymbol{W}}_2 \hat{\boldsymbol{W}}_1 \hat{\boldsymbol{x}}) \boldsymbol{x}^\top \boldsymbol{W}_1^\top \right\} \\
&= \boldsymbol{P}_1 \boldsymbol{W}_1 \boldsymbol{W}_1^\top - \hat{\boldsymbol{W}}_2 \hat{\boldsymbol{W}}_1 \text{diag}(\boldsymbol{I}; \boldsymbol{0}) \boldsymbol{W}_1^\top \\
&= \boldsymbol{P}_1 (\boldsymbol{U}\boldsymbol{U}^\top + \boldsymbol{V}\boldsymbol{V}^\top) - \boldsymbol{W}_2 \boldsymbol{U}\boldsymbol{U}^\top
\end{aligned}$$

$$\begin{aligned}
\nabla_{P_2}\mathcal{L} &= \mathbb{E}_{\boldsymbol{x},\hat{\boldsymbol{x}}} \left\{ (\boldsymbol{P}_2 \boldsymbol{W}_2 \boldsymbol{W}_1 \boldsymbol{x} - \hat{\boldsymbol{W}}_1 \hat{\boldsymbol{x}}) \boldsymbol{x}^\top \boldsymbol{W}_1^\top \boldsymbol{W}_2^\top \right\} \\
&= \boldsymbol{P}_2 \boldsymbol{W}_2 \boldsymbol{W}_1 \boldsymbol{W}_1^\top \boldsymbol{W}_2^\top - \hat{\boldsymbol{W}}_1 \text{diag}(\boldsymbol{I}; \boldsymbol{0}) \boldsymbol{W}_1^\top \boldsymbol{W}_2^\top \\
&= \boldsymbol{P}_2 \boldsymbol{W}_2 (\boldsymbol{U}\boldsymbol{U}^\top + \boldsymbol{V}\boldsymbol{V}^\top) \boldsymbol{W}_2^\top - \boldsymbol{U}\boldsymbol{U}^\top \boldsymbol{W}_2^\top
\end{aligned}$$

$$\begin{aligned}
\nabla_{W_1}\mathcal{L} &= \mathbb{E}_{\boldsymbol{x},\hat{\boldsymbol{x}}} \{ \boldsymbol{P}_1^\top (\boldsymbol{P}_1 \boldsymbol{W}_1 \boldsymbol{x} - \hat{\boldsymbol{W}}_2 \hat{\boldsymbol{W}}_1 \hat{\boldsymbol{x}}) \boldsymbol{x}^\top + \\
&\quad + \boldsymbol{W}_2^\top \boldsymbol{P}_2^\top (\boldsymbol{P}_2 \boldsymbol{W}_2 \boldsymbol{W}_1 \boldsymbol{x} - \hat{\boldsymbol{W}}_1 \hat{\boldsymbol{x}}) \boldsymbol{x}^\top \} \\
&= \boldsymbol{P}_1^\top \boldsymbol{P}_1 \boldsymbol{W}_1 - \boldsymbol{P}_1^\top \hat{\boldsymbol{W}}_2 \hat{\boldsymbol{W}}_1 \text{diag}(\boldsymbol{I}; \boldsymbol{0}) + \\
&\quad + \boldsymbol{W}_2^\top \boldsymbol{P}_2^\top \boldsymbol{P}_2 \boldsymbol{W}_2 \boldsymbol{W}_1 - \boldsymbol{W}_2^\top \boldsymbol{P}_2^\top \hat{\boldsymbol{W}}_1 \text{diag}(\boldsymbol{I}; \boldsymbol{0}) \\
&= \boldsymbol{P}_1^\top \boldsymbol{P}_1 \boldsymbol{W}_1 - \boldsymbol{P}_1^\top \boldsymbol{W}_2 [\boldsymbol{U}; \boldsymbol{0}] + \boldsymbol{W}_2^\top \boldsymbol{P}_2^\top \boldsymbol{P}_2 \boldsymbol{W}_2 \boldsymbol{W}_1 - \boldsymbol{W}_2^\top \boldsymbol{P}_2^\top [\boldsymbol{U}; \boldsymbol{0}]
\end{aligned}$$

$$\begin{aligned}
\nabla_{W_2}\mathcal{L} &= \mathbb{E}_{\boldsymbol{x},\hat{\boldsymbol{x}}} \{ \boldsymbol{P}_2^\top (\boldsymbol{P}_2 \boldsymbol{W}_2 \boldsymbol{W}_1 \boldsymbol{x} - \hat{\boldsymbol{W}}_1 \hat{\boldsymbol{x}}) \boldsymbol{x}^\top \boldsymbol{W}_1^\top \} \\
&= \boldsymbol{P}_2^\top \boldsymbol{P}_2 \boldsymbol{W}_2 \boldsymbol{W}_1 \boldsymbol{W}_1^\top - \boldsymbol{P}_2^\top \hat{\boldsymbol{W}}_1 \text{diag}(\boldsymbol{I}; \boldsymbol{0}) \boldsymbol{W}_1^\top \\
&= \boldsymbol{P}_2^\top \boldsymbol{P}_2 \boldsymbol{W}_2 (\boldsymbol{U}\boldsymbol{U}^\top + \boldsymbol{V}\boldsymbol{V}^\top) - \boldsymbol{P}_2^\top \boldsymbol{U}\boldsymbol{U}^\top
\end{aligned}$$

From $\nabla_{W_1}\mathcal{L}$, we get

$$\nabla_U\mathcal{L} = \boldsymbol{P}_1^\top \boldsymbol{P}_1 \boldsymbol{U} - \boldsymbol{P}_1^\top \boldsymbol{W}_2 \boldsymbol{U} + \boldsymbol{W}_2^\top \boldsymbol{P}_2^\top \boldsymbol{P}_2 \boldsymbol{W}_2 \boldsymbol{U} - \boldsymbol{W}_2^\top \boldsymbol{P}_2^\top \boldsymbol{U}$$

$$\nabla_V\mathcal{L} = \boldsymbol{P}_1^\top \boldsymbol{P}_1 \boldsymbol{V} + \boldsymbol{W}_2^\top \boldsymbol{P}_2^\top \boldsymbol{P}_2 \boldsymbol{W}_2 \boldsymbol{V}$$

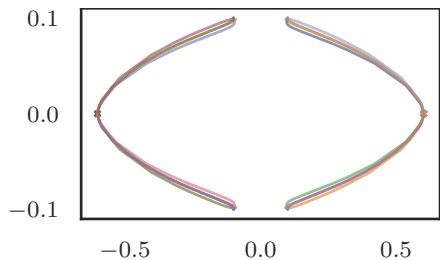

Figure 3: **Differential equation trajectories**

The SGD iterations are then

$$\boldsymbol{W}_1^{(t+1)} - \boldsymbol{W}_1^{(t)} = -\eta_w \nabla_{W_1} \mathcal{L}$$
$$\boldsymbol{W}_2^{(t+1)} - \boldsymbol{W}_2^{(t)} = -\eta_w \nabla_{W_2} \mathcal{L}$$
$$\boldsymbol{P}_1^{(t+1)} - \boldsymbol{P}_1^{(t)} = -\eta_p \nabla_{P_1} \mathcal{L}$$
$$\boldsymbol{P}_2^{(t+1)} - \boldsymbol{P}_2^{(t)} = -\eta_p \nabla_{P_2} \mathcal{L}$$

Clearly, the above SGD iterations correspond to highly non-linear differential equation systems in high dimensional space. If we do not make further simplifications, there is little we can do analytically. To get a deeper insight , we consider a special case where all matrices are diagonal ones. This simplification allows us to discuss the dynamics analytically. To ensure that the discussion under such simplification can generalize to the original problem, we verify our main conclusion via numerical simulation.

To continuous our discussion, we assume that $d = 2m$, $\boldsymbol{U} = \mathrm{diag}(u^{(1)}, u^{(2)}, \cdots, u^{(m)})$, $\boldsymbol{W}_2 = \mathrm{diag}(w_2^{(1)}, w_2^{(2)}, \cdots, w_2^{(m)})$, $\boldsymbol{P}_1 = \mathrm{diag}(p_1^{(1)}, p_1^{(2)}, \cdots, p_1^{(m)})$, $\boldsymbol{P}_2 = \mathrm{diag}(p_2^{(1)}, p_2^{(2)}, \cdots, p_2^{(m)})$. To avoid notation clutter, the superscript is omitted in the following discussion when it is clear from the context.

The differential equations for the above SGD iterations are

$$\dot{u} = \eta_w (p_1 w_2 - p_1^2 + p_2 w_2 - p_2^2 w_2^2) u$$
$$\dot{v} = -\eta_w (p_1^2 + p_2^2 w_2^2) v$$
$$\dot{w}_2 = -\eta_w (u^2 + v^2) p_2^2 w_2 + \eta_w p_2 u^2$$
$$\dot{p}_1 = -\eta_p (u^2 + v^2) p_1 + \eta_p u^2 w_2$$
$$\dot{p}_2 = -\eta_p (u^2 + v^2) w_2^2 p_2 + \eta_p u^2 w_2$$

Again, these equations are high-order ones. We use numerical simulation to plot their trajectories. We initialized all variables in $\{\pm \xi\}$ where $\xi = 0.1$ is a small number. The SGD learning rate $\eta_w = 0.1$, $\eta_p = 1.0$. Since there are 5 variables (including $\{u, v\}$), we try all the possible initialization combinations and obtain 32 trajectories in Figure 3. The x-axis is $u(t)$ and the y-axis is $v(t)$ The marker X indicates the termination point of the numerical simulation. It is easy to see that all 32 trajectories converge to $\lim_{t \to \infty} v(t) = 0$ and $\lim_{t \to \infty} u(t) = \pm 0.5 \neq 0$. This means that the differential equations indeed converge to the ideal solution where the coefficient of the augmentation subspace converges to zero and the coefficient of the invariant subspace converges to a non-zero fixed point.

## A.2 HCCL WITH MOMENTUM ENCODER

For a fair comparison with BYOL, we also evaluated HCCL with momentum encoder. In this experiment, HCCL has the same architecture as Figure 1, but the parameters of the two branches are no longer shared. As in BYOL, the online network(the branch with predictor) is defined by a set of weights $\theta$, and is comprised of three stages: an encoder $f_\theta$, a hierarchical projection head $g_\theta^n$ and $n$ predictor $h_1, h_2, ..., h_n$. The target network(the branch without predictor) uses a different set of

Table 8: **Top-1 accuracy under linear evaluation on ImageNet.** '†' denotes improved reproduction from SimSiam.

| Method | Encoder | 100ep | 200ep | 400ep | 800ep |
|---|---|---|---|---|---|
| BYOL† | R50 | 66.5 | 70.6 | 73.2 | 74.3 |
| **HCCL + Momentum Encoder** | **R50** | **70.9** | **72.8** | **74.2** | **74.9** |

weights $\xi$, and is comprised of two stages: an encoder $f_\xi$, a hierarchical projection head $g_\xi$. For encoder and projection head, the target parameters $\xi$ are an exponential moving average of the online parameters $\theta$. Given a target decay rate $\tau \in [0, 1]$, after each training step we perform the following update:

$$\xi = \tau\xi + (1 - \tau)\theta$$

We set $\tau = 0.996$, other training settings are the same as in section 4. Limited by training resources, we give the results under 100, 200, 400, 800 epoch in Table 8. Compared with BYOL, HCCL with momentum encoder is significantly improved.

### A.3 MEMORY FOOTPRINT AND TRAINING SPEED.

We compared the memory footprint and speed of SimSiam, BYOL, and HCCL during training. All experiments are trained on 8 x V100 GPU with batchsize of 512 and mixed precision. Table 9 shows that HCCL hardly increases memory footprint and training time compared to SimSiam and BYOL.

### A.4 DIFFERENT WAY OF CROSS-CORRELATION.

We further study the performance of establishing cross-correlation relationships in different ways. When the level of the hierarchical projection head increases, the cross-correlation combination between different levels will become more complicated. In this subsection, we explored the cross-correlation combination based on the 3 levels hierarchical projection heads. Table 10 shows that cross-correlation combinations between different levels can help improve performance. With more levels of hierarchical projection architecture, the cross-correlation method will become very complicated and cannot be verified one by one through experiments. We believe that the more cross-level cross-correlation relationships are established, the better the performance will be. We will explore this in future work.

### A.5 ANALYZE OF HIERARCHICAL FEATURE.

In this subsection, we analyze the features extracted by the hierarchical projection head to further explore the mystery behind HCCL.

#### A.5.1 **Similarity between features of different levels of hierarchical projection head.**

Using SimSiam and HCCL, we extracted the features of 50000 images on the Imagenet Val dataset. Each image is randomly augmented to obtain two views. For SimSiam, each image can extract

Table 9: **Memory footprint and Training speed.** 'Memory' refers to the memory footprint on a single gpu during training. 'ME' means momentum encoder.

| Method | Encoder | Memory | Speed |
|---|---|---|---|
| SimSiam | R50 | 8220M | 1290 im/s |
| BYOL | R50 | 8400M | 975 im/s |
| HCCL | R50 | 8320M | 1220 im/s |
| HCCL + ME | R50 | 8660M | 940 im/s |

Table 10: **Top-1 accuracy under different way of cross-correlation.** All method use 3 level hierarchical projection head.

| Method | Top-1 Acc |
|---|---|
| a | 68.4 |
| b | 68.7 |
| c | 69.2 |

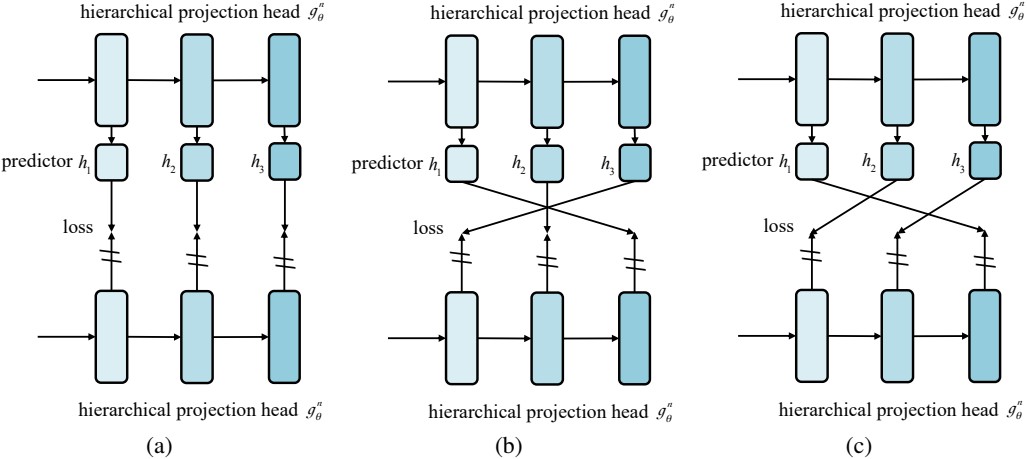

Figure 4: **Different way of cross-correlation on 3 level hierarchical projection head.** '=' denotes stop gradient.

two features, and we calculate the similarity (cosine distance) of the two features. For HCCL, we use a 2-level hierarchical projection head, so each image can extract four features, including two high-level projectors features and two low-level projector features. We calculate the similarity of high-level features, the similarity of low-level features, and the similarity between high-level and low-level features. Figure 5 shows the cosine distance distribution of all samples in the data set. We observed that the features extracted in the same layer of the projector are very similar in both HCCL and SimSiam, and most of their cosine distances are less than 0.2. For HCCL, there are large differences between low-level and high-level projector features. This is consistent with our view that different layers of the projector extract different information. Intuitively, contrastive learning between too similar features is not a good choice.

### A.5.2 **Feature similarity within and between classes.**

We extracted the hierarchical projector features of 50000 images on the Imagenet Val dataset. Each sample extracts low-level and high-level features respectively. For each class, we calculate the average cosine distance between all samples in the class, and the results are shown in Figure 6(a). Then we calculate the average cosine distance between the samples of this class and the samples of other classes, the results are shown in Figure 6(b). We can observe that the average cosine distance within classes is smaller than that between classes, whether it is low-level or high-level projector features. The difference is that the average cosine distance of high-level projector features is greater than that of low-level projector features, both within and between classes. We believe that the high-level projector makes semantic disturbance to the low-level features, making the distribution of features more divergent. HCCL's hierarchical projection head introduces diversity to the features, which is equivalent to different 'data augmentation' at the feature level. Finally, We use cross contrast loss to learn invariance in these features that experience different disturbances.

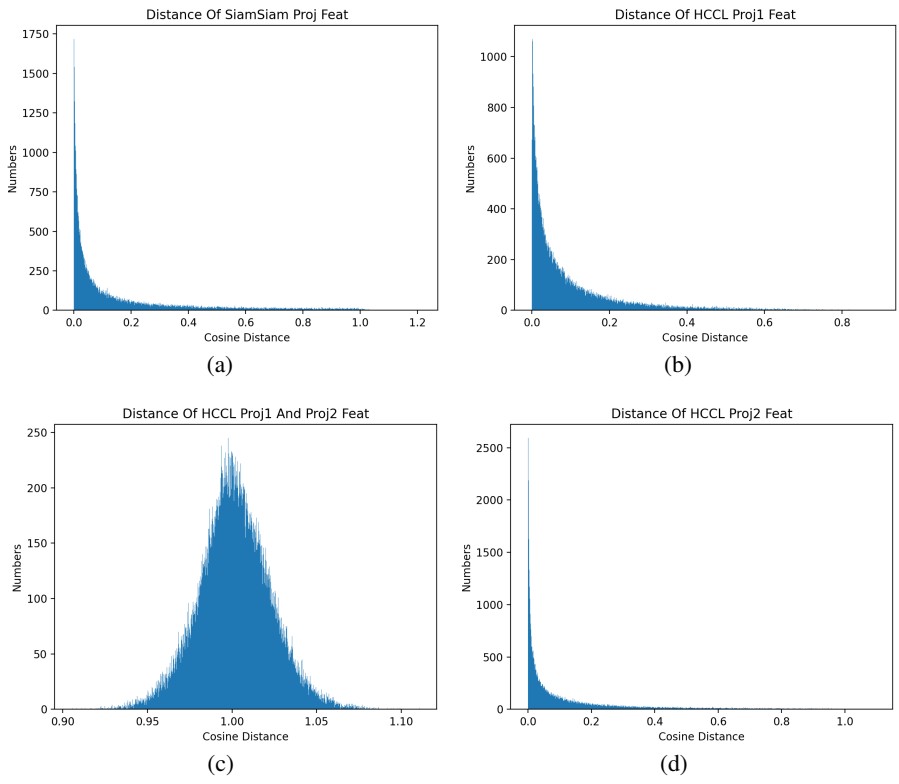

Figure 5: **Cosine distance distribution of view1 and view2 features.** (a): SimSiam's projector features. (b) HCCL's low level projector features. (c) HCCL's low level and hightlevel projector features. (d) HCCL's high level projector features.

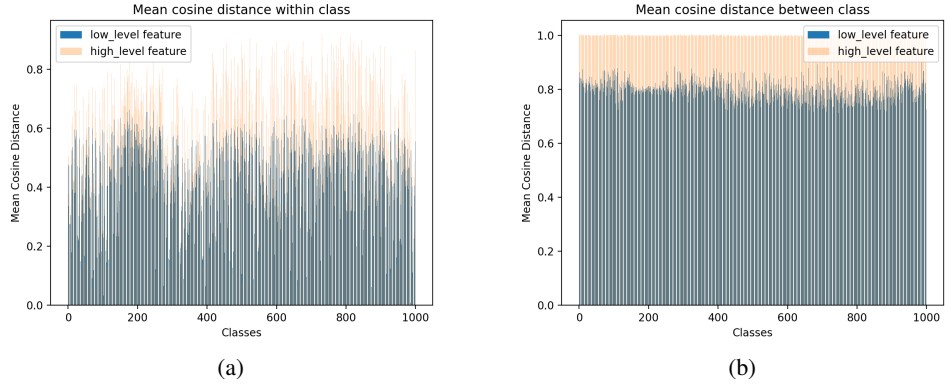

Figure 6: **Average cosine distance within and between classes.** (a): Average cosine distance of features in each class. (b): Average cosine distance of features between different class.

