# OpenReview forum: "Hierarchical Cross Contrastive Learning of Visual Representations"
_ICLR.cc/2022/Conference — ICLR 2022 Submitted_

### Official Review · Reviewer_mW6T · 2021-11-02

**Correctness:** 3
**Technical Novelty And Significance:** 3
**Empirical Novelty And Significance:** 2
**Recommendation:** 6
**Confidence:** 5

**Details Of Ethics Concerns:**

None.

**Main Review:**

Strengths
1. The proposed approach, especially the cross contrastive loss, is novel and interesting.
2. The proposed approach is fairly easy to implement.
3. The proposed approach is evaluated on multiple benchmarks and achieves noticeable improvements compared with previous states of the art.

Weaknesses
1. Although empirically evaluated, the intuition behind the cross contrastive loss is still not clear. More theoretical analysis or interpretation should be provided to understand why cross contrastive loss works better. Specifically, what is the difference in features learned with and without cross contrastive loss? Will the features across different projector layers become closer after trained with cross contrastive loss?
2. The training protocol for serval downstream tasks is not consistent with previous work. Since the authors simply report the performance from existing literature, it would be more critical to use consistent training protocols for a rigorous comparison. For example,
    - Table 2 in this manuscript reported the results of Table 1 in Barlow Twins [1]. However, Barlow Twins uses a learning rate of 0.002 for the conv-net weights and 0.5 for the final linear layer, while HCCL uses 0.02 and 300, respectively.
    - Similarly, Table 3 reported the results of Table 3 in Barlow Twins. But the training epochs for Places-205 is 14 for Barlow Twins, but 30 for HCCL.
3. On page 6, the authors pointed out that "BYOL has a greater training cost, because It[it] requires 4 times forward in one iteration." However, HCCL has much more trainable parameters by introducing more projectors and predictors. Taking two projector heads as example, HCCL has about 35% more parameters compared with BYOL (~49M for HCCL and ~36M for BYOL). It would more reasonable to compare the memory footprint and time during training instead of forward-propagation times.
4. In Table 7, the "Fixed" learning rate is confusing. Does it mean the predictors us the same learning rate as the rest of the network?
5. In Table 8, what is the performance of HCCL+Momentum Encode for 400ep and 800ep? Can it outperform BYOL? If not, what is the possible reason that HCCL cannot benefit more from longer training time compared with BYOL and Barlow Twins?
6. There are several typos in the manuscript:
   - In the caption of Figure 1, ’̸=’ should be `=’;
   - Page 6, "because It requires 4 times..." should be "because it requires 4 times...";
   - Page 7, 2nd line, missing space in "Table2";
   - Page 8, 3rd paragraph, missing space in "section4".

[1] Jure Zbontar, Li Jing, Ishan Misra, Yann LeCun, and Ste ́phane Deny. Barlow twins: Self-supervised learning via redundancy reduction. arXiv preprint arXiv:2103.03230, 2021.

**Summary Of The Paper:**

This manuscript proposed a new contrastive self-supervised learning approach (HCCL). Compared with exiting work such as BYOL and SimSiam, HCCL introduced (1) multiple hierarchical projectors and predictors; and (2) contrastive loss calculated across different layers of projectors. HCCL is empirically evaluated on several self-supervised learning benchmarks (e.g., iNat18, Place-205, and COCO instance seg, etc.) and achieves noticeable improvement compared with previous states of the art (e.g., SWAV, BYOL, SimSiam, and Barlow Twins). Additionally, ablation studies are provided to verify the significance of hierarchical projectors, cross contrastive loss, and higher learning rates of the predictor.

**Summary Of The Review:**

The authors proposed an interesting hierarchical comtrastive learning approach that achieves noticeable improvement over serval benchmarks.

However, it would be more convincing if some theoretical analysis or empirical interpretation could be provided to help the readers understand why hierarchical project heads and cross contrastive loss works better. Additionally, some justification regarding downstream task training protocols should be provided for the sake of fair and rigorous comparison with previous work.

---

> ### Author Response · Authors · 2021-11-21
> **For Reviewer mW6T**
>
> Thank you for the constructive comments.
>
> ## Question 1
> We think that the features extracted from the same layer of projector contain relatively similar information,
> but the features extracted from different layers of the projector contain more different information.
> The purpose of using cross-contrast loss is to build a learning relationship between features that are more different.
> We believe that the features learned with cross-contrast loss will be more robust in semantics.
> And the features across different projector layers become closer after being trained with cross contrastive loss.
> But compared to the feature of the same projector layer, they still maintain a greater distance.
> We have added some analysis of the feature of the hierarchical projection head in Appendix A.5 of the paper.
>
> ## Question 2
> We agree with your point of view, but currently, different work uses different protocols, it is difficult to find a standard protocol to strictly compare. Here are some examples：
>
> ### Semi-supervised training on ImageNet:
>
> * SWAV: For 1% finetuning, they use a learning rate of 0.02 for the trunk and 5 for the final layer. For 10% finetuning, they use a learning rate of 0.01 for the trunk and 0.2 for the final layer.
>
> * BYOL： Sweep over the learning rate {0.01, 0.02, 0.05, 0.1, 0.005} and the number of epochs {30, 50} and
> select the hyperparameters achieving the best performance on their local validation set to report test performance
>
> * SimCLR: They use a learning rate of 0.8 without warmup. For 1% labeled data they fine-tune for 60 epochs, and for 10% labeled data they fine-tune for 30 epochs.
>
> * BT: They train for 20 epochs with a learning rate of 0.002 for the ResNet-50 and 0.5 for the final classification layer.
>
> ### Transfer learning on Places205:
> * SWAV: lr 0.01, epochs=28
>
> * BT: lr 0.01, epochs=14
>
> We try to ensure the fairness of the training epoch.
>
> ## Question 3
> We compared the memory footprint and training speed of HCCL, BYOL, and SimSiam during training, and the results are shown in appendix A.3 of the paper. Although more projection heads and predictors are introduced, HCCL hardly brings more training time and memory consumption.
>
> ## Question 4
> 'Fixed' means that during training, the learning rate of the predictor is always fixed, but the learning rate of the rest of the network will decrease in a cosine manner.
>
> ## Question 5
> Yes, HCCL + momentum encoder outperforms BYOL in 400ep and 800ep. The experimental results have been added to Appendix A.2 of the paper.
>
> ## Question 6
> Thank you for the detailed comments, We have revised the paper.

---

> > ### Comment · Reviewer_mW6T · 2021-11-28
> > **The Severe Conflict between Method Intuition/Explanation and Experimental Evaluation in Fig. 5**
> >
> > Thanks for your answers to Questions 3-6. I also partially agree with the authors' augment for Question 2.
> >
> > However, for Question 1, the experimental results in Appendix A.5 are conflicted with the intuition/explanation of HCCL.
> >
> > First of all, in Fig. 5 (c), I do not understand why Cosine distance can be larger than one, whose range should be exact [0, 1]. This result makes me worry about the rigorousness of this paper.
> >
> > Second, even without the first concern, the experimental results still cannot explain HCCL. As written in the abstract, "By cross-level contrastive learning, HCCL ...... regulates invariant ...... crosses different levels." However, Fig. 5 (c) shows that the HCCL-trained features are still significantly different across different levels (Cosine distance ~ 1.0), demonstrating that HCCL can NOT "regulate invariant crosses different levels" at all. This severe conflict makes me worry about the mechanism of HCCL.

---

> > > ### Author Response · Authors · 2021-11-28
> > > **For Reviewer mW6T**
> > >
> > > Thank you for your Comment.
> > >
> > > (1) We use cosine distance, not cosine similarity. Cosine distance = 1- cosine similarity, and the range of cosine distance is 0 ~ 2.
> > >
> > > (2) The features of different levels of projection head are quite different. Although HCCL can regulate invariant crosses different levels, it can not bring them close to a very similar degree.
> > >
> > > In addition, the contrast loss does not directly affect the projection head features. The predictor transforms the projection head features, so the projection head features are not very similar.
> > >
> > > Finally, we also compare the cosine distances of different levels of projection head features in Figure 2 (b)(Hierarchical projection head without cross contrastive loss). Their average cosine distance is greater than HCCL(1.121>0.994), which indicates that the cross contrast loss is regular invariant crosses different levels.

---

> > > > ### Comment · Reviewer_mW6T · 2021-11-29
> > > > **HCCL features across different levels have cosine similarity equal to 0.006.**
> > > >
> > > > First of all, thanks for your clarification of the Cosine distance calculation.
> > > > However, my second concern is not addressed yet.
> > > >
> > > > 1. Since Cosine distance = 1- cosine similarity, the cosine similarity of features trained w/ and w/o cross contrastive loss are 0.006 and -0.121, respective. Both of them have cosine similarity fairly close to 0, meaning the features cross different layers are orthogonal or decorrelated with each other. I still do not believe it can support the authors' claim. At least the "regularization" is not very effective. I do not think "the contrast loss does not directly affect the projection head features" is a strong argument either, since the Cosine distance of features within the same layer mostly falls in the range [0, 0.2], which is also affected by the predictor.
> > > > 2. Besides, even the difference between the Cosine distance is relatively small (0.994 vs. 1.121), which may not be statistically meaningful.
> > > > 3. Last but not least, considering the Figure 2 network only has a 2-level hierarchical projection head instead of 3 levels as used in the main paper, the regularization power may be further reduced, especially between features from the 1st and 3rd layers.

---

> > > > > ### Author Response · Authors · 2021-11-30
> > > > > **For Reviewer mW6T**
> > > > >
> > > > > Thanks for your comments.
> > > > >
> > > > > (1) We calculated the average cosine distance (0.129) between the high-level predictor features and the low-level projection features, which is similar to the average cosine distance between the high-level projection features (0.133, Figure 5(d)). This means that HCCL narrows the distance between different levels of projection features (0.994 to 0.129) by using predictor. (Applying cross-contrast loss directly to the projection feature will cause collapse)
> > > > >
> > > > > The features of the same level of the projection head are extracted by the same network structure and have a high degree of similarity regardless of whether they are trained or not. On the contrary, features of different levels are extracted by different structures, and their distributions are quite different. The purpose of HCCL is to learn the invariance that exists in different distributions, but it does not guarantee that the features of different distributions will always become very similar. The invariance between them may be tiny, but it can also bring benefits.
> > > > >
> > > > > (2) We believe that this difference has a certain significance under the statistics of a large amount of data. For example, whether it is SimSiam, BYOL, or HCCL, training 800 epoch compared to training 100 epoch, the average cosine similarity of the two views in the entire training set may only increase by 0.02, but the top-1 accuracy of ImageNet can increase by 3% to 4%.
> > > > >
> > > > > (3) We used 2-level hierarchical projection heads in both the main paper and Figure 2. You can refer to Sec 3.3 Architecture part (page 4) and Sec 5.1 (page 8)

---

> > > > > > ### Comment · Reviewer_mW6T · 2021-11-30
> > > > > > **Final Rating: Marginally below the acceptance threshold**
> > > > > >
> > > > > > I appreciate the quick responses and detailed explanations regarding all my questions.
> > > > > >
> > > > > > Overall, I think the idea of using cross contrastive loss with hierarchical projection heads is theoretically sound and interesting.
> > > > > > The experiments results over multiple benchmarks also show improvements.
> > > > > >
> > > > > > However, I think the actual contributions and the real mechanism behind HCCL are questionable, which is also pointed out by reviewer #TpLR and #NpsJ. Especially, I still think the HCCL mechanism can not be fully explained by the 0.006 cosine similarity between the high-level projection features and the low-level projection features (it is sort of trivial to say "cosine distance between the high-level predictor features and the low-level projection features is low" since the model is trained in this way).
> > > > > >
> > > > > > In sum, I would give "Marginally below the acceptance threshold" as my final rating.

---

### Official Review · Reviewer_TpLR · 2021-11-02

**Correctness:** 3
**Technical Novelty And Significance:** 3
**Empirical Novelty And Significance:** 2
**Recommendation:** 6
**Confidence:** 3

**Main Review:**

The consistency assumption over the representation in different views and different layers has been partially explored in many feature regularization approaches. In most cases, this assumption was directly implemented with a consistency loss between feature pairs.  While the consistency across views is well established as the basis of contrastive learning, most practices showed that the features are only partially similar among the last few layers. For this reason, researchers have to carefully select the layers and parameters in tuning the performance. see examples of related work in [1-4].

The authors address the limitation of the assumption above with several transformation modules between compared features. The performance did show good improvement in the cases of limited training data and short training epochs. The improvement for the full training sets and full training epochs are, however, limited. This may lead to the concern that the information introduced by the proposed approach has been naturally learned with the existing datasets and training settings.

Reference:
[1] Hao Guo, Kang Zheng, Xiaochuan Fan, Hongkai Yu, and Song Wang. Visual attention consistency under image transforms for multi-label image classification. CVPR, 2019.
[2] Yuenan Hou, Zheng Ma, Chunxiao Liu, and Chen Change Loy. Learning lightweight lane detection CNN by self-attention distillation. ICCV, pages 1013–1021, 2019.
[3] Lezi Wang, ZiyanWu, Srikrishna Karanam, Kuan-Chuan Peng, Rajat Vikram Singh, Bo Liu, and Dimitris N Metaxas. Sharpen focus: Learning with attention separability and consistency. ICCV, 2019.
[4] Zeyi Huang, Yang Zou, Vijayakumar Bhagavatula, Dong Huang, Comprehensive Attention Self-Distillationfor Weakly-Supervised Object Detection, NeurIPS, 2020.

**Summary Of The Paper:**

The authors propose an extension to the contrastive learning based representation learning approach. The proposed method, call Hierarchical Cross Contrastive Learning(HCCL), leverages the features in different levels and views for more consistent features over the standard CL using only the features of the final layers. The effectiveness is validated in classification, detection, segmentation, and few-shot learning tasks.

**Summary Of The Review:**

The approach proposed in this paper is straightforward and well implemented. The DNNs trained by this approach benefit mostly in the case of small training sets and epochs. The advancement over the full training settings is still small due to the natural limitation of the consistency assumption mentioned above.

---

> ### Author Response · Authors · 2021-11-21
> **For Reviewer TpLR**
>
> Thank you for the constructive comments.
> The references you mentioned are very helpful to us, we cite them in the paper.
>
> ## Why the improvement for the full training sets and full training epochs are limited?
> * HCCL's hierarchical projection head introduces diversity to the features, which is equivalent to different 'data augmentation' at the feature level.
> However, when there are enough training samples, the features are already diverse enough, and the diversity brought by the augmentation of the features is not significant enough, so the performance improvement is limited
> * The significance of our work is to propose a simple and scalable method to improve the performance of contrastive learning.
> HCCL can be easily combined with other contrastive learning frameworks.
> Although compared with other methods, the performance improvement of HCCL is not particularly significant, but this improvement is almost completely costless.
> * We have supplemented the experimental results of HCCL combined with BYOL in Appendix A.2 of the paper,
> and supplemented the memory usage and speed during training in Appendix A.3.

---

### Official Review · Reviewer_NpsJ · 2021-11-02

**Correctness:** 3
**Technical Novelty And Significance:** 2
**Empirical Novelty And Significance:** 2
**Recommendation:** 5
**Confidence:** 4

**Main Review:**

Strenghs
- The paper writing is clear.
- The cross-level contrastive learning shows the improvements in comparison to previous methods.


Weaknesses

In general, the contributions of the paper is incremental from previous works as most of the building blocks of the proposed method are adopted from SimSiam (Chen & He, 2021) and BYOL (Grill et al. 2020).

1. Novelty: the main difference between the proposed and SimSiam approaches are the hierarchical projection heads and the cross contrastive loss between each head in a branch with the head of previous level in the other branch.
Although the experimental results show some improvements, the intuition of cross contrastive loss to make improvement is unclear.
For example, as the encoder f already extract high-level features, which information is further embedded in the hierarchical projection heads?
Which aspect (i.e. feature distributions, embedded information, etc.) can be enhanced in cross contrastive loss to help the improvements?

2. Experiments: As illustrated in ablation study (Table 5, 6, 7), the improvements seem to be the results of several tuning steps (i.e. number of additional levels, predictor learning rate). In particular, the adaptation of Hierarchical and Contrastive loss only cannot help to improve the performance as presented in Table 5.
The authors should analyze in details to show the actual improvements obtained by the proposed approach.


**Summary Of The Paper:**

In this paper, the authors proposed a Hierarchical Cross Contrastive Learning (HCCL) method for Self-supervised learning (SSL) of visual representation.
The proposed method include a design of a hierarchical projection network that produces multi-level latent representations. A cross contrastive loss is also introduced to learn invariant visual representations.
HCCL is validated on several downstream tasks including classification, segmentation and object detection.

**Summary Of The Review:**

Although the experimental results show some improvements, the contributions of the paper is incremental from previous works and the intuition for the improvement is unclear.

---

> ### Author Response · Authors · 2021-11-21
> **For Reviewer NpsJ**
>
> Thank you for your insightful comments.
>
> ## What is the novelty of HCCL?
> * HCCL is a simple and universal method that does not depend on a specific architecture and can be combined with many contrastive learning methods.
> * The advantage of HCCL is that it can improve the performance of other methods with almost no additional consumption.
> It is a good choice when training resources are limited.
> * We have supplemented the experimental results of hccl combined with BYOL in Appendix A.2 of the paper,
> and supplemented the memory footprint and speed during training in Appendix A.3.
>
> ## The intuition of cross contrastive loss
> In contrast learning, the encoder extracts high-order features, and the projection head will further eliminate redundant information in the high-order features, and only retain invariance.
> On this basis, we construct the cross-contrastive loss mainly based on two reasons:
>
> * The information extracted from different layers of the network is different, that is, there are certain differences in the features extracted from different layers of the projection head.
> (Information Bottleneck Theory [1])
> * Increasing the difference between contrast features helps to improve performance.
> (BYOL uses a momentum encoder to make the difference, and its performance is better than simsiam.
> NNCLR[2] uses the nearest neighbor feature to make the difference.)
>
> Therefore, we use hierarchical projection heads to extract features of different levels and construct cross-contrastive loss between features of different levels.
> To a certain extent, the hierarchical projection head is equivalent to further augmentation of higher-order features, and the cross-contrast loss will aggregate these augmented features.
> We have analyzed the features extracted by the hierarchical projection head in Appendix A.5.
> The statistical results show that the features of different levels are very different.
> This difference enriches the distribution of features and helps improve learning performance.
> However, it is difficult to explain what specific information is embedded in the layered projection head.
> As far as we know, the current research on the projection layer is not perfect, and we hope to study further in the future.
>
> ## Experiments
> We thank you for pointing out this issue. We have added an experiment in the paper, that is,'Deep projection with multi predictor'.
> The structure diagram is in Figure 2(e), and the experimental results are in Table 5.
> The experimental results show that only using multi predictors can not improve performance.
> In fact, Hierarchical, cross-contrast loss, and multi predictor are indispensable in HCCL.
> Hierarchical ensures that differentiated features of different layers can be extracted,
> cross-contrast loss ensures that learning can be performed between differentiated features,
> and multi predictor ensures that the unique information of different layers of features will not be destroyed.
>
> [1] Tishby N . The information bottleneck method[J].  1999.
>
> [2] Dwibedi, D. , et al. "With a Little Help from My Friends: Nearest-Neighbor Contrastive Learning of Visual Representations." (2021).

---

### Official Review · Reviewer_RrYz · 2021-11-03

**Correctness:** 4
**Technical Novelty And Significance:** 3
**Empirical Novelty And Significance:** 3
**Recommendation:** 6
**Confidence:** 4

**Main Review:**

This paper starts with the motivation of self-supervised learning methods followed by the summarisation of the important methods. Afterward, the paper observes the trend of the existing methods being imag-level predictions utilizing global representations. Paper argues that the gap between the image-level pre-training and downstream tasks is not filled yet. It sounds interesting but it is not clearly mentioned how this gap exists in a more detailed manner.  As this paper advocates the need for hierarchical cross contrastive learning, a clear insight on the issue with the methods relying only on a global representation learning would make this paper stronger.

The method is straight forward and it is clearly present in the paper. Pseudocode has made even easier to understand the pipeline.

Another strong point about this paper is the extensive evaluations. Although the performance improvement over the existing methods is marginal in most of the cases, the pair have done evaluations on multiple benchmarks and also present the ablation studies.



**Summary Of The Paper:**

This paper presents a hierarchical cross contrastive self-supervised learning framework for learning visual representation. This paper proposes to project the representations of an image and its augmented version to multiple latent spaces and also make predictions on each of the latent spaces.  A contrastive loss between the features of different projection levels is minimized to learn the parameters. Experiments on the image classification and detection benchmarks are evaluated. A comparison between important existing methods is also done in the paper.

**Summary Of The Review:**

The research problem is interesting.  Novelty on the idea is modest and the performance compared with the existing method is also modest.

The presentation of the paper is good but it lacks a clear insight on the motivation for hierarchical architecture and also the way cross-correlation is established. What would be the performance when cross-correlation has been established in a different manner than that present in the paper? As there is potential to have different combinations of cross-correlation dependencies between the layers.

---

> ### Author Response · Authors · 2021-11-21
> **For Reviewer RrYz**
>
> Thank you for your insightful comments.
>
> ## The improvements of HCCL seem small, what is its advantage?
> * The advantage of HCCL is that it is simple and versatile, and can be easily combined with other comparative learning frameworks.
> * Although compared with other methods, the performance improvement of HCCL is not particularly significant,
> but this improvement is almost completely costless. And HCCL has more obvious advantages when the training epoch is small.
> In the case of limited training resources, HCCL is a better choice.
> * We have supplemented the experimental results of HCCL combined with BYOL in Appendix A.2 of the paper,
> and supplemented the memory usage and speed during training in Appendix A.3.
>
> ## What is the motivation for hierarchical architecture?
> * We observe that BYOL outperforms SimSiam by using the momentum encoder.
> So we intuitively believe that the momentum encoder adds a small semantic perturbation to the feature, which makes the features of the two views more different.
> This greater differentiation can help learn more general features.
> * Furthermore, the information bottleneck theory shows that the network extracts different information at different layers.
> It means that in contrastive learning, the features of two views extracted from the same layer contain relatively similar information, but the features extracted from different layers contain more different information.
> Naturally, we built a hierarchical structure, and cross-contrastive loss is established on the representations of different layers. Utilizing the differences in features of different layers, we can achieve similar performance without using the momentum encoder.
> * Our method can also be easily extended to different contrastive learning works. We have added some analysis of the representation of the hierarchical projection head in Appendix A.5 of the paper.
>
> ## How cross-correlation is established?
> * We insist on the view that it is best to establish a cross-correlation relationship between different levels.
> Since we found that the two-level hierarchical projection head has the best effect, we only show the different ways of establishing cross-correlation under the two-level hierarchical projection head architecture.
> * For your questions, in In Appendix A.4 of the paper, we added the experimental results of different ways of cross-correlation under the three-level hierarchical projection head architecture.
> The experiment still shows that when all the cross-correlations are established between different levels, it achieves the best performance.
> * With more levels of hierarchical projection architecture, the cross-correlation method will become very complicated and cannot be verified one by one through experiments.
> We believe that the more cross-level cross-correlation relationships are established, the better the performance will be.
> We will explore this in future work.

---

> > ### Comment · Reviewer_RrYz · 2021-12-01
> > **thank you**
> >
> > I appreciate the response from the authors.

---

### Decision · Program_Chairs · 2022-01-20

**Decision:**

Reject

**Comment:**

In general, the reviewers were lukewarm about the paper. They all acknowledged the strength of the paper: it is well written, HCCL showed (somewhat) improvements over previous methods, and it is easy to implement. However, it still feels incremental, and the improvement over the full training setting is small due to the natural limitation of consistency assumption. The AC feels that while there is merit of the proposed method, the impact seems to be limited to specific scenarios such as limited epochs.